# Association of Polymorphism *CHRNA5* and *CHRNA3* Gene in People Addicted to Nicotine

**DOI:** 10.3390/ijerph191710478

**Published:** 2022-08-23

**Authors:** Krzysztof Chmielowiec, Jolanta Chmielowiec, Aleksandra Strońska-Pluta, Grzegorz Trybek, Małgorzata Śmiarowska, Aleksandra Suchanecka, Grzegorz Woźniak, Aleksandra Jaroń, Anna Grzywacz

**Affiliations:** 1Department of Hygiene and Epidemiology, Collegium Medicum, University of Zielona Góra, 28 Zyty St., 65-046 Zielona Gora, Poland; 2Independent Laboratory of Health Promotion, Pomeranian Medical University in Szczecin, 70-204 Szczecin, Poland; 3Department of Oral Surgery, Pomeranian Medical University in Szczecin, 72 Powstanców Wlkp. St., 70-111 Szczecin, Poland; 4Department of Pharmacokinetics and Therapeutic Drug Monitoring, Pomeranian Medical University, 70-111 Szczecin, Poland; 5Private Dental Practice, 9 Bahnhofstrasse, 3940 Steg, Switzerland

**Keywords:** gene, *CHRNA5*, *CHRNA3*, rs16969968, rs578776, rs1051730, nicotine addiction

## Abstract

Smoking is a chronic and relapsing addictive trait that harms public health. Among the many identified genetic variants of nicotine dependence, the variants in the *CHRNA5/A3/B4* gene cluster on chromosome 15 that encode the α5, α3, and β4 subunits have recently received a lot of attention. Importantly, variants in this gene cluster have been associated with nicotine addiction. Among the many significant variants in this cluster, the polymorphism SNP rs16969968 seems to be the most interesting factor in nicotine addiction. This polymorphism causes an amino acid change from aspartate to asparagine at position 398 of the α5 nicotinic receptor protein sequence. Our study aimed to analyze three polymorphic variants: the rs16969968 located in the *CHRNA5* gene, the rs578776 and rs1051730 located in the *CHRNA3* gene in nicotine-addicted subjects, and in controls. Our study encompasses an association analysis of genotypes and haplotypes. A group of 401 volunteers was recruited for the study and divided into two groups: the study group consisted of addicted smokers and a control group of 200 unrelated non-smokers who were not dependent on any substance and healthy. A statistically significant difference was observed in the frequency of genotypes of the rs1051730 polymorphism of the *CHRNA3* gene (χ^2^ = 6.704 *p* = 0.035). The T/T genotype was statistically significantly more frequent in the group of nicotine-dependent subjects. The haplotypes rs16969968, rs578776, and rs1051730 were distinguished, of which the G-T-T and G-C-T haplotypes were present only in the study group. With differences in frequencies, statistical significance was noted—for the G-T-T haplotype *p* = 0.01284 and the G-C-T haplotype *p* = 0.00775. The research stated that novel haplotypes G-T-T and G-C-T, though with very low-frequency variants in *CHRNA3*, were associated with nicotine addiction.

## 1. Introduction

Smoking is a chronic and relapsing addictive trait that harms public health. In addition to the proven impact of smoking on the risk of cardiovascular and respiratory diseases and malignant neoplasms, epidemiological studies show a significant effect of smoking on the condition of the oral cavity and the development of periodontal diseases. According to statistics from the World Health Organization [1], smoking kills about six million people worldwide annually. Over five million deaths are caused by direct cigarette smoking, and approximately 600,000 people die from secondary or passive exposure to nicotine smoke. If these trends continue, it is likely that by 2030 the number of deaths among smokers will increase to over 8 million per year [2].

Cigarette smoke contains about 4000 compounds, yet nicotine is the main substance responsible for developing addiction. Nicotine acts mainly in the brain via neural nicotinic acetylcholine receptors (nAChRs). They are widespread in the central and peripheral nervous systems. They are also called gated ion channel ligands, consisting of five subunits that span the membrane. They can modulate the release of neurotransmitters, which include dopamine, glutamate, and GABA. It also mediates signal transmission in synapses.

There are 12 nAChR neuronal subunits: 9 α (α2–α10) and 3 β (β2–β4) [3,4,5]. These subunits organize themselves into numerous distinct pentametric nAChRs, which result in the formation of receptors that differ in their distribution in the body and biological and other pharmacological properties [6]. The interaction of nicotine with the nAChR forms the molecular basis of the reward provided by nicotine and the development of nicotine addiction. Variants nAChRs are not only a potential risk factor for nicotine addiction but also a target for smoking cessation efforts and personalized medicine to treat nicotine addiction and other mental disorders.

Growing evidence indicates that nicotine and cigarette smoke alter the expression of small regulatory molecules 21–24 nt long, called microRNAs (miRNAs). MiRNAs are predicted to regulate most mammalian protein-encoding genes [7,8,9]. These regulatory molecules are particularly abundant in the brain and play an important role in several aspects of nervous system development [10,11,12]. A subset of miRNAs regulates gene expression via functional interactions with miR-NA recognition elements (MREs) in the 3′ UTRs of mammalian nAChR receptor subunit genes. Furthermore, several miRNAs are regulated by chronic nicotine exposure in the mammalian brain.

Many twin studies show that, along with environmental factors, nicotine dependence is caused by genetic variation, estimated heredity of around 50%. Many studies have been conducted to identify susceptibility loci and genetic variants for nicotine dependence and associated phenotypes, such as genome-wide linkage analysis, candidate gene association, and GWAS. Among the many identified genetic variants of nicotine dependence, the variants in the *CHRNA5*/*A3/B4* gene cluster on chromosome 15 (Figure 1) that encode the α5, α3, and β4 subunits have recently received a lot of attention [13,14,15,16]. Importantly, variants in this gene cluster have been associated with nicotine addiction and lung cancer [17,18,19]. As a consequence of these genetic studies, new efforts were made to understand how genetic variants in this region influence nicotine addiction and related phenotypes at the molecular level. The reproducibility of the research showing the genetic link between the variants in the *CHRNA5*/*A3/B4* gene cluster and nicotine addiction further increases the power of these findings. Among the many significant variants in this cluster, the polymorphism SNP rs16969968 (Figure 2) seems to be the most interesting factor in nicotine addiction. This polymorphism causes an amino acid change from aspartate to asparagine at position 398 of the α5 nicotinic receptor protein sequence.

Of course, the very mechanism of action of clustered nAChR subunits in the development of nicotine addiction is not fully understood. Still, we can cite animal studies and evidence-based on mouse knockout (KO) or mutation models of nAChR subunits, especially the α5 subunit, suggesting that the disruption of α5 * nAChR signaling significantly reduces the stimulating effects of nicotine on the mHb-IPN pathway, and thus enables the consumption of larger amounts of nicotine [20]. This is why variants in the *CHRNA5*/*A3/B4* cluster play a role in nicotine addiction through the aversive effect of nicotine on the mHb-IPN pathway when there are few reports of nicotine enhancement in the ventral tegmental (VTA) DA neurons [21,22]. The *A3/B4* gene cluster and addiction to nicotine (but also to other substances) is a complex and challenging phenotype. This phenotype has many symptoms, including smoking early in the morning, increasing smoking, and greater tolerance and ease of relapse after quitting smoking. It is worth emphasizing that the development of nicotine addiction does not happen suddenly. This starts with experimental smoking with the first puff to smoke regularly and, finally, the effect of nicotine addiction [23]. Up to now, the *CHRNA* haplotypes have been less studied than their genotypes (gene and genome-wide association studies, GWAS) and their direct implication in human brain nicotine-related phenotypes [24]. There are some publications in the literature on the protective role of 4-SNP haplotypes indicating that the rs4887074 might be associated with nicotine dependence as it may modulate the effect of rs16969968 in GWAS datasets. At the same time, the G alleles of rs16969968 and rs4887074 (associated with *CHRNA5* and *CHRNB4,* respectively) placed on common haplotypes seem to protect and interact with each other [25].

There are many tools for assessing nicotine addiction. The Fagerström Test and the Diagnostic and Statistical Manual for Mental Disorders (4th edition; DSM-IV) are the most common tests for evaluating the severity of nicotine addiction. However, there is only a limited correlation between the two measures highlighting different aspects of nicotine addiction [26,27]. Compared to the DSM-IV, the Fagerström test is a simplified measure. In particular, it pays attention to the number of cigarettes smoked a day and the time between waking up and the first cigarette. The DSM-IV, on the other hand, pays attention to the behavioral and emotional aspects of addiction.

Our study aimed to analyze three polymorphic variants, i.e., the rs16969968 located in the *CHRNA5* gene, the rs578776, and rs1051730 located in the *CHRNA3* gene in nicotine-addicted subjects and controls. Our study encompasses an association analysis of genotypes and haplotypes. 

## 2. Materials and Methods

### 2.1. Participants

The study was conducted in the Independent Laboratory of Health Promotion, Pomeranian Medical University in Szczecin. A group of 401 volunteers was recruited for the study and divided into two groups: the study group consisted of 201 addicted smokers (M = 35, SD = 14.06, minimum age 20, maximum age 78) and a control group of 200 unrelated non-smokers who were not dependent to any substance and healthy (M = 25, SD = 13.12, minimum age 18, maximum age 72). During the recruitment, the study participants were asked to fill out a Fagerstrom test. The study was conducted in accordance with the Declaration of Helsinki principles and approved by the Ethics Committee. All subjects signed their informed consent for participating in the research. No financial or other compensation for participating in the study was granted. 

### 2.2. DNA Isolation and Genotyping

The isolation of genomic DNA was carried out following standard procedures and the recommendation of the reagent manufacturer. After DNA isolation, appropriate dilutions were prepared per the standards for real-time PCR reactions. The real-time polymerase chain reaction method using the LightCycler^®^ 480 II system was used to analyze the polymorphic sites of the selected genes, and genotyping was performed using the difference in melting points of individual nucleotides.

### 2.3. Statistical Analysis

The Hardy-Weinberg equilibrium and the differences in frequency of genotypes and alleles between nicotine-dependent (ND) smokers and non-smokers were analyzed by the chi-square test using the Statistica 13 software (Tibco Software Inc., Palo Alto, CA, USA) for Windows (Microsoft Corporation, Redmond, WA, USA). The R version 1.2 program (modules genetics and haplo stats) was used to analyze the linkage disequilibrium and the frequency of individual haplotypes.

## 3. Results

The frequency distributions accorded with the HWE in the study group for polymorphism rs16969968 located in *CHRNA5* and rs578776 located in *CHRNA3* (Table 1). No statistically significant differences were observed in the frequencies of genotypes and alleles of the polymorphic variants of the rs16969968 between nicotine-dependent cases and non-smokers (Table 2).

The frequency distributions accorded with the HWE in the study group for polymorphism rs1051730 located in *CHRNA3* (Table 1). As shown in Table 2, a statistically significant difference was observed in the frequency of genotypes of the rs1051730 polymorphism of the *CHRNA3* gene (χ^2^ = 6.704 *p* = 0.035). The T/T genotype was statistically significantly more frequent in the group of nicotine-dependent subjects (n = 37, 18.41%). A statistically significant difference was also observed in the frequency of alleles of the rs1051730 polymorphism of the *CHRNA3* gene (χ^2^ = 4.650 *p* = 0.031). The T allele was statistically significantly more frequent in the group of smokers (n = 159, 39.55%).

In the presented analysis, the linkage disequilibrium between analyzed polymorphisms was proved statistically significant (Table 3).

Due to the relatively small studied group in the haplotype analysis, the entire group was not divided into homogeneous subgroups. The research was performed only for juxtaposing haplotypes composed of the three polymorphisms studied. As shown in Table 4, five haplotypes were distinguished, of which the G-T-T and G-C-T haplotypes were present only in the study group. With differences in frequencies, statistical significance was noted—for the G-T-T haplotype *p* = 0.01284 and the G-C-T haplotype *p* = 0.00775.

The results of the Fagerstrom questionnaire were analyzed only in nicotine-dependent subjects. This group was divided into females and males, and the analyses were carried out for each group divided by sex. There were significant differences in Fagerstrom test results between smoking men and women (*p* = 0.0456) (Table 5).

## 4. Discussion

In 2006, McClure and Swan [28] claimed that with the help of phenotypic and genetic data, it is possible to explain and predict the risk of addiction. Some researchers described the genetic mapping of nicotine addiction as a very complex formula [29]. The individual genes influence the development of the different stages of addiction [30]. On the other side, the “mature” phenotype of addiction can be defined by many correlated functions and traits [31,32] related to genetic associations [33]. 

The specificity in the genotypic mapping of the smoking individuals is the complexity and multidimensionality of their endophenotypes spectrum [34]. An excellent example of such analysis is the study carried out in 2005 by Cannon et al. [35] concerning the taste-sensory subscale dedicated to nicotine-addicted subjects. Baker et al. [34] proposed a schematic model of nicotine addiction which explains the addiction phenotype using three basic criteria: (1) pattern of heavy, continuous, and “automatic” tobacco use; (2) relapse after trying to quit smoking; and (3) the severity of the recall. Each underlying criterion has been theoretically and empirically related to the global dependency construction [36], with a slight overlap between these functions [37] and various factors [38].

Our study described phenotypes in the addicted to nicotine and control groups, those who were related to oral cavity health. The analysis concerned selected indicators and their relationships with specific genotypes of the selected polymorphic variants. However, the primary phenotypic characteristic was the division of all the individuals into smokers and non-smokers. Next, the smokers were analyzed concerning their gender and nicotine addiction intercourse using the Fagerström’s Nicotine Dependency Test [27], a standard tool for assessing the intensity of physical addiction to nicotine. It contains six items that assess the number of cigarettes one smokes, the compulsion to use, and addiction. The analyses were carried out in two different homogeneous gender subgroups to estimate the biological aspects of addiction. In our study result, significant differences were found between smoking men and women, showing that smoking males were more highly addicted indeed. (*p* = 0.0456, Table 5). In the preliminary study in adolescents who were children of alcoholics, the predictive models of drug use were found [39]. It assumes that hypodopaminergic functioning may predict drug use in males, while a deleterious environment seems to be the salient predictor in females [39]. In animal models of nicotine/tobacco addiction, similar trends were observed. The preference in females and adolescents appears to be more pronounced than in males and adults. On the other hand, the decline in smoking in developed countries is observed, expressed more in women than in men [40]. Social factors may impact smoking in humans. The data from nonhuman subjects in plenty of controlled experiments showed that gender differences in nicotine/tobacco addiction have a biological basis. The effect of gonadal hormones may underline only part of the gender differences observed in the nicotine abusers, although knowledge along with gender is a potential factor that seems important in an individual smoking cessation program [41]. Many social factors may influence the gender differences in smoking, such as alcohol abuse, being prone to gambling, eating disturbances, mood and emotional hesitance, kind of job, self-awareness, self-care, and so on, which are commonly known.

The emergence and consolidation of tobacco dependence are related to the variability of the smoking person’s behavior over time, which has been termed the “smoking career”. Low FTND scores are classified as having a low nicotine dependence. This suggests that they may not need Nicotine Replacement Therapy (NRT), although it is recommended that they still be monitored for withdrawal symptoms. 

Many researchers have shown a link between *CHRNA* genes and nicotine addiction. The studies described the complexity of the phenotype and its influence on the course of addiction. 

When considering nicotine addiction in the context of a multi-gene and multi-factor disease entity, but taking into account the genes analyzed in this study, the works of Weiss and colleagues from 2008 [16] should be cited as analyzing the problem in its very essence. Most importantly, these studies show a significant proportion of the correlation between genes and interaction with the environment. The most significant associations were found between the *CHRNA5*-A3-B4 variants in nicotine-addicted subjects who started smoking relatively early—e.g., at 16 years of age. Then such a phenomenon was called the early onset of nicotine addiction. Interestingly, the lack of association has been demonstrated in late-onset smokers with haplotypes in the *CHRNA5*-A3-B4 cluster. Second, the association results showed a significant odds ratio, almost a two-fold increase. And third, the research is notable because the same association pattern was observed in three large groups of over two thousand independent samples.

The reported results were confirmed by other researchers who analyzed the same gene cluster. In a study of Israeli women, Greenbaum et al. in 2006 obtained results indicating a clear relationship between their measure of dependence and the *CHRNA3* SNP rs1051730 [42,43]. Sherva et al. in 2008 also described a significant relationship between rs16969968 and smoking status, as well as experiencing “pleasant noise” in response to early smoking experiments [44].

In our study, a statistically significant difference was observed in the frequency of genotypes of the rs1051730 polymorphism of the *CHRNA3* gene in smokers and non-smokers. The T/T genotype was significantly more frequent in the group of smokers. A significant difference was also observed in the frequency of alleles of the rs1051730 polymorphism of the *CHRNA3* gene in both groups. The T allele was significantly more frequent in the group of smokers. 

Subsequent studies also provided evidence that the *CHRNA5*-A3-B4 cluster is a candidate gene and found evidence for associations between common variants in the cluster and their respective phenotypes [14,45]. Three new genome-wide association studies have also shown strong evidence for an association between *CHRNA5*-A3-B4 variants and lung cancer, but they vary by connection [17,18,43].

To fully understand the molecular mechanism related to the *CHRNA5*-A3-B4 gene cluster with nicotine addiction and lung cancer, it is necessary to determine which SNP can be described as changing the biological function. It seems that the most transparent SNP is rs16969968: it probably contributes biologically to the development of nicotine addiction as it turns an amino acid into an α5 nicotinic receptor protein. This change is in the adjacent large cytoplasmic domain to the conserved amphipathic α-helix, so it is far from extracellular acetylcholine in its binding site and is unlikely to affect agonist binding sensitivity. In such a region, negatively charged Asp398 can promote Ca^2+^ permeability, while Asn398, replaced with an amide group instead of a negatively charged carboxyl group, can inhibit it. Consistent with this hypothesis, recent studies have shown that the D398N polymorphism affects the function of (α4β2) 2α5 nAChR [13]. 

The association of the rs16969968 genotype with smoking is consistent with the studies by Hong et al. [46]. They showed that the rs16969968 genotype significantly explains 3.3% and 4.6% of the variance in the severity of nicotine addiction and the number of cigarettes smoked per day, respectively [46]. The “gene exposure” effect of vmPFC (Ventromedial prefrontal cortex) is consistent with a meta-analysis [47] of neuroimaging pharmacology studies that showed that smoking and CHRNA agonist administration in adult smokers are associated with lower neuronal activity in, among others, vmPFC. Consistent with these findings, a further meta-analysis examining the neurobiological targets of pharmacological and cognitive treatments for nicotine addiction has shown that similar parts of vmPFC have lower activity in smokers [48]. Moreover, the smoking interaction rs16969968 provides evidence that nicotine, and not other chemicals in cigarettes, may be the basis of the relationship between smoking and GMV reduction modulated by the nicotinic acetylcholine receptor system [49]. In the Bangladeshi population, on the other hand, the following SNPs: rs16969968-rs578776 and rs578776-rs11072768, turned out to be in a linkage disequilibrium (LD), which indicates a high probability of being together and then transferred to later generations [50].

Another interesting study found that the 3′-UTR rs578776 variant, which causes changes in the regulatory sequence of the *CHRNA3* gene, is associated with behaviors related to smoking nicotine, and the mutated homozygous genotype A/A protects against smoking intensification [14,43,45,51,52]. The GG/AA genotype combination for the rs16969968/rs578776 variants provided the greatest “protection” against smoking, while AA/GG was associated with the highest risk of smoking [47,48,49]. On the other hand, rs11072768 is a variant of the intron, which turned out to be significantly related to the number of cigarettes smoked during the day, and smoking more for a long time in Chinese men [53,54,55,56]. In Korean men, the G allele in rs11072768 *CHRNB4* was associated with the onset of smoking, the amount of smoking, and cessation of smoking [44,57]. This study showed a stronger association of the rs16969968 and rs578776 polymorphisms with a positive SS. The rs16969968 polymorphism was revealed as a risk factor for SS in the Bengali population, although this finding was not statistically significant. The polymorphic A allele rs16969968 was also found predominantly in smokers compared to non-smokers. Bierut et al. [13] and Kuryatov et al. [58] say carriers of the A allele may express nAChR subtypes of α5 subunits with diminished function, consequently increasing the rate of cigarette consumption. This, in turn, may explain the high prevalence of polymorphic rs16969968 SNPs in smokers.

On the contrary, the rs578776 polymorphism had a protective effect on positive SS, and the rs578776 polymorphic allele A was the highest among non-smokers, confirmed by several reports on the relationship between this allele and SS [13,14,44,45,51,52,59,60,61,62,63,64,65,66,67,68,69,70,71,72]. Hong et al. [73] report a connection between the rs578776 G allele and activation of a specific brain circuit, which may explain the previously described studies. In the case of the rs11072768 polymorphism, no association with SS was found. In conclusion, the combination of GG/AA genotypes may contribute to negative SS, while AA/GG may have the greatest risk of smoking.

The cited results also identified being in the group of students as a significant risk factor for the rs16969968 polymorphism, but statistically insignificant after Bonferroni correction, and being in the group of 25–50 year-olds as a protective factor for the rs578776 polymorphism. As most volunteers were in the 25–50 age group, the presence of rs16969968 and the absence of rs578776 variants may increase the risk of smoking in young adults. Increased transmission of the rs578776 A allele has also been reported in those with a history of tobacco exposure from their father. This leads to a possible dependency that the A allele may be part of a combination of SNPs in strong linkage disequilibrium with each other and are therefore transferred in families. Family studies are needed to elucidate this risk allele’s exact transmission mechanism. Additionally, two new haplotypes (A-A-T and A-A-G) were observed in addition to the six haplotypes described by the 1000 Genomes Project; among them, the GAG haplotype was reported as a protective haplotype even after Bonferroni correction [50].

However, each study did not clearly and with certainty determine the quantitative influence of genetic variants on the phenotype of nicotine dependency. It is still a complex and unsolved problem. In our study, the search was crowned with the definition of characteristic haplotypes. Such an analysis was carried out and presented along with determining the linkage between the selected SNPs presented in the paper. Due to the relatively small studied group in the haplotype analysis, the entire group was not divided into homogeneous subgroups. The research was performed only for juxtaposing haplotypes composed of the three SNPs studied. Different haplotypes were distinguished, of which the G-T-T and G-C-T haplotypes were present only in the study group (they did not appear in the control group in any of the cases). Statistical significance was noted with differences in frequencies—for the G-T-T haplotype and the G-C-T haplotype. This result should be interpreted with great caution—although significant, but not define the haplotype characteristic of nicotine addiction. However, this analysis is promising as replication in a larger group may yield important results.

## 5. Conclusions

Summing up, we are critical of the presented results. Disregarding the numerous significant results, we obtained an ambiguous answer to the questions posed. The research stated that novel haplotypes G-T-T and G-C-T, though with very low-frequency variants in CHRNA3, were associated with nicotine addiction.

However, it gives hope that the research direction is proper and that the analyses should be conducted in subsequent studies.

## Figures and Tables

**Figure 1 ijerph-19-10478-f001:**
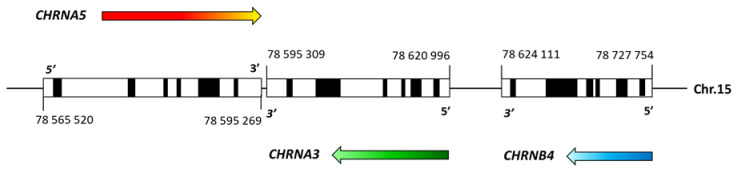
Diagram of the human *CHRNA5*/*A3/B4* gene cluster. Each gene is drawn to scale, with black boxes representing exons and white boxes representing untranslated regions. Colored arrows indicate the direction of transcription. The locations of each gene are marked. Homo sapiens chromosome 15, GRCh38.p14 Primary Assembly; NCBI Reference Sequence: NC_000015.10 (https://www.ncbi.nlm.nih.gov/nuccore/NC_000015.10 (accessed on 15 June 2022)).

**Figure 2 ijerph-19-10478-f002:**
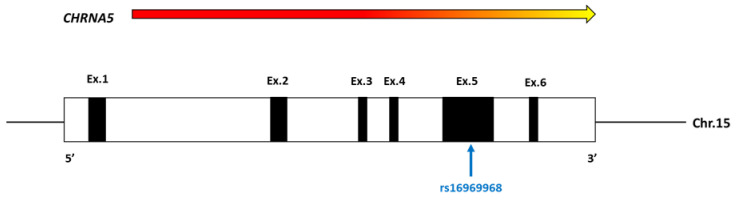
Scheme of the *CHRNA5* gene. The black rectangles represent the exons, and the white rectangles represent the untranslated regions. A colored arrow indicates the direction of transcription.

**Table 1 ijerph-19-10478-t001:** Hardy–Weinberg equilibrium of the *CHRNA5* rs16969968, rs578776 *CHRNA3,* and rs1051730 *CHRNA3* in the group of nicotine-dependent (ND) subjects and controls.

		Observed (Expected)		χ^2^	*p* Value
rs16969968 *CHRNA5*					
Controls	G/G	90 (91.1)	A allele freq = 0.33 G allele freq = 0.67	0.131	>0.05
	A/G	90 (87.8)			
	A/A	20 (21.1)			
ND	G/G	85 (81.5)	A allele freq = 0.36 G allele freq = 0.64	1.131	>0.05
	A/G	86 (93.0)			
	A/A	30 (26.5)			
rs578776 *CHRNA3*					
Controls	C/C	108 (108.8)	T allele freq = 0.26 C allele freq = 0.74	0.081	>0.05
	C/T	79 (77.4)			
	T/T	13 (13.8)			
ND	C/C	113 (112.7)	T allele freq = 0.25 C allele freq = 0.75	0.014	>0.05
	C/T	75 (75.6)			
	T/T	13 (12.7)			
rs1051730 *CHRNA3*					
Controls	G/G	90 (91.8)	T allele freq = 0.32 C allele freq = 0.68	0.340	>0.05
	A/G	91 (87.4)			
	A/A	29 (20.8)			
ND	G/G	85 (81.5)	T allele freq = 0.36 C allele freq = 0.64	0.131	>0.05
	A/G	86 (93.0)			
	A/A	30 (26.5)			

**Table 2 ijerph-19-10478-t002:** Frequencies of genotypes and alleles of *CHRNA5* rs16969968, rs578776 *CHRNA3*, and rs1051730 *CHRNA3* in the group of nicotine-dependent (ND) subjects and controls.

	Controls	ND	Test χ^2^ (*p* Value)
rs16969968 *CHRNA5*			
G/G (%)	90 (45.00%)	85 (42.29%)	2.231 (0.328)
A/G (%)	90 (45.00%)	86 (42.79%)	
A/A (%)	20 (10.00%)	30 (14.92%)	
G (%)	270 (67.50%)	256 (63.68%)	1.300 (0.255)
A (%)	130 (32.50%)	146 (26.32%)	
rs578776 *CHRNA3*			
C/C (%)	108 (54.00%)	113 (56.22%)	0.215 (0.898)
C/T (%)	79 (39.50%)	75 (37.31%)	
T/T (%)	13 (6.50%)	13 (6.5%)	
C (%)	295 (73.75%)	301 (74.88%)	0.130 (0.715)
T (%)	105 (25.25%)	101 (25.22%)	
rs1051730 *CHRNA3*			
C/C (%)	90 (45.00%)	79 (39.30%)	6.704 (0.035) *
C/T (%)	91 (45.50%)	85 (42.23%)	
T/T (%)	19 (9.50%)	37 (18.41%)	
C (%)	271 (67.75%)	243 (60.45%)	4.650 (0.031) *
T (%)	129 (32.25%)	159 (39.55%)	

*—statistically significant results.

**Table 3 ijerph-19-10478-t003:** Linkage disequilibrium between polymorphisms rs16969968, rs578776, and rs1051730 rs1051730 in the group of nicotine-dependent (ND) subjects and controls.

	rs578776	rs1051730
rs16969968 D’ *p* value n	0.9992 <2.2204 × 10^−16^ * 400	0.9941 <2.2204 × 10^−16^ * 400
rs578776 D’ *p* value n		0.8743730 <2.2204 × 10^−16^ * 400

n—number of cases; D’—standardized Lewotin disequilibrium factor; *p*—statistical significance; *—statistically significant results.

**Table 4 ijerph-19-10478-t004:** Haplotypes frequencies of rs16969968, rs578776, and rs1051730 in the group of nicotine-dependent (ND) subjects and controls.

Haplotype	Hap-Score	*p*	Frequency	Controls	ND
G-C-C	−1.3084	0.19074	0.39026	0.4125	0.3682
G-T-C	−0.8508	0.39488	0.24937	0.2625	0.2363
A-C-T	1.2014	0.22957	0.34287	0.3225	0.3632
G-T-T	2.4615	0.01384 *	0.00749	NA	0.0149
G-C-T	2.6627	0.00775 *	0.00874	NA	0.0174

*p*—statistical significance; *—statistically significant results.

**Table 5 ijerph-19-10478-t005:** Mann-Whitney U test of the Fagerstrom questionnaire in female and male smokers.

n Females	n Males	M Females	M Males	SD Females	SD Males	Z	*p*
105	96	3.35	4.11	2.59	2.58	−1.9987	0.0456 *****

*p*-statistical significance with the Mann–Whitney U-test; n—number of subjects; M ± SD—mean ± standard deviation. * statistically significant results.

## Data Availability

Not applicable.

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
