# Peer review of "Association of Polymorphism *CHRNA5* and *CHRNA3* Gene in People Addicted to Nicotine"

_ijerph, 2022, doi:10.3390/ijerph191710478_

Round 1

Reviewer 1 Report

This large scale study of nicotine addiction genotypes is generally well written and the tables and graphs are nicely displayed. The HWE tests are not significant. Some minor comments are Table 3 LD 0,87 should be 0.87, and Table 4 should consider Fisher's exact test. 

With that said, however, there are several substantive concerns the authors should address. 

1. It appears that this study was part of a larger study on oral health. That needs to be described. The study for the current paper seems to be a secondary data analysis rather than specifically designed to test the current hypotheses. 

2. The mean FTND scores are low, raising questions on whether all smokers are actually addicted. A FTND score of 1 to 2 indicates a very low level of addiction and some would argue that less than 5 cigarettes per day is nonaddicting. As the authors note, there are many measures of dependence including cravings, withdrawal symptoms, cessation success that are not measured by FTND. This needs to be considered in the discussion. 

3. It is unclear what is meant in the abstract conclusion that the results were not unambiguous. Some findings were significant and others weren't. That is the case with most studies, genetic or otherwise. 

4. It is not entirely clear what are the novel aspects of the study. It seems to be on CHRNA haplotypes and this should be emphasized more in the abstract and introduction unless I am misunderstanding this paper. As such, while CHRNA haplotypes have been less studies than genotypes, there are some publications in the literature (lee et al. 2018. combined genetic influences....). These should be mentioned, and the current results compared to other work.  

5. The discussion section is very broad in scope, especially the first paragraphs. Nicotine addiction is too large a literature to summarize in this paper. The discussion should be pared back, and focused on the current findings. Similarly, the conclusion needs to be reworded. It seem like the authors are calling their own data invalid. Or are they?, in which case this argues against publication. If the argument is that CHRNA snps are not associated with dependence despite expectations than that need to be stated. Or to me, at least, some snps/haploypes are and some snps are not. What is new?

Author Response

Dear Reviewer,

Thank you very much for your review and valuable comments. We analyzed all the comments and replied to each, indicating where and how the corrections in the Manuscript were made, indicating the line and page.

Below are the point-by-point answers.

With respect

Authors

This large scale study of nicotine addiction genotypes is generally well written and the tables and graphs are nicely displayed. The HWE tests are not significant. Some minor comments are Table 3 LD 0,87 should be 0.87, and Table 4 should consider Fisher's exact test.

With that said, however, there are several substantive concerns the authors should address.

  1. It appears that this study was part of a larger study on oral health. That needs to be described. The study for the current paper seems to be a secondary data analysis rather than specifically designed to test the current hypotheses.

Thank you for this suggestion. True, the study was part of a larger study that looked at the genetic factors predisposing people to nicotine addiction and the effects of smoking on oral health. This article only describes the results that relate genetic factors to nicotine addiction. In order not to raise doubts about the test by the recipient, the fragment concerning the examination of the oral health condition has been removed line 145-149 page 4.

  1. The mean FTND scores are low, raising questions on whether all smokers are actually addicted. A FTND score of 1 to 2 indicates a very low level of addiction and some would argue that less than 5 cigarettes per day is nonaddicting. As the authors note, there are many measures of dependence including cravings, withdrawal symptoms, cessation success that are not measured by FTND. This needs to be considered in the discussion.

Thank you very much for this suggestion. The Fagerström's Nicotine Dependency Test is a standard tool for assessing the intensity of physical addiction to nicotine. The test was designed to provide an ordinal measure of the nicotine addiction associated with smoking. It contains six items that assess the number of cigarettes you smoke, the compulsion to use, and addiction. The emergence and consolidation of tobacco dependence are related to the variability of the smoking person's behavior over time, which has been termed the "smoking career". Low FTND scores are classified as having a low nicotine dependence. This suggests that they may not need Nicotine Replacement Therapy (NRT), although it is recommended that they still be monitored for withdrawal symptoms. We have covered this in the discussion on line 233-258 page 7.

  1. It is unclear what is meant in the abstract conclusion that the results were not unambiguous. Some findings were significant and others weren't. That is the case with most studies, genetic or otherwise.

Thank you for your suggestion us to check and correct this part of the article. We tried to do our best to make it more transparent and correspond with the facts of the case. This sentence before does not make sense.

Line: 34-37, page 1; Disregarding the numerous significant results, we did not obtain an unambiguous answer to the questions posed.

It could sound now this way:
To assume the gathered results may be preliminary report that the nicotine- dependent subjects may have their nicotinic acetylcholine receptor more sensitive to addiction behavior not only because of more frequent changes in CHRNA3 gene but also T/T genotypes haplotype. These data support that both CHRNA3 and CHRNA5 gene polymorphisms are the candidates in nicotine addiction especially that their role is still unclear even regards the preclinical (now these are lines: 37-39,
page 1).

  1. It is not entirely clear what are the novel aspects of the study. It seems to be on CHRNA haplotypes and this should be emphasized more in the abstract and introduction unless I am misunderstanding this paper. As such, while CHRNA haplotypes have been less studies than genotypes, there are some publications in the literature (lee et al. 2018. combined genetic influences....). These should be mentioned, and the current results compared to other work.

Thank you for suggestion. You are right, of course, that the role of haplotypes should be more explained and emphasized in an introduction, as it was unclear why we analyzed them in this paper. The disproportional place was given to GWAS than to haplotypes. We tried to fulfill it now, referring to a few other publications. (lines: 118 – 125, page 3).

5.The discussion section is very broad in scope, especially the first paragraphs. Nicotine addiction is too large a literature to summarize in this paper. The discussion should be pared back, and focused on the current findings.

Thank you for your doubts about the main ideas the authors should have expressed in the discussion. We tried to verify this part of the dissertation and make it much shorter. We modified the order of presented results a little, giving ours at the beginning and other authors in father parts of the discussion to make it more focused on the current finding. We hope it will be acceptable now.

Similarly, the conclusion needs to be reworded. It seem like the authors are calling their own data invalid. Or are they?, in which case this argues against publication. If the argument is that CHRNA snps are not associated with dependence despite expectations than that need to be stated. Or to me, at least, some snps/haploypes are and some snps are not. What is new?

Thank You for this conclusion. We analyzed the collected data once again and tried to formulate such new summing up:

To assume the gathered results may be preliminary report that the nicotine- dependent subjects may have their nicotinic acetylcholine receptor more sensitive to addiction behavior not only because of more frequent changes in T/T genotypes in CHRNA3 gene but also G-T-T and G-C-T haplotypes. These data support that CHRNA3 and CHRNA5 gene polymorphisms are the strong candidates in nicotine addiction unless their role is still unclear even regards the preclinical models. However, it gives hope that the direction of the research is proper and that the analyses should be conducted continuously.

Reviewer 2 Report

This is an interesting and brave manuscript where the authors addressed a complex issue involving both genetic and sociological aspects, namely- the relevance of specific polymorphisms in the nicotinic acetylcholine receptors 3 and 5 for the tendency to get addicted to smoking.  The authors reached a well-balanced and careful summary inferring that they are on the right way, but cannot offer a definitive conclusion. This exceptional summation is appreciated and is a positive aspect of this study.

However, the manuscript also includes negative aspects which are weak points and may require corrections in the revised text, as is listed below.

First and foremost, the authors refer to SNPs as if these are the only elements that make a difference in the susceptibility for addiction, while small non-coding RNAs may contribute significantly to the studied phenomenon and were not even mentioned, although others have repeatedly showed significant impact of cholinergic genes-targeting  microRNAs and transfer RNA fragments on diverse related phenomena.

Second, the issues of sex and gender are presented vaguely, and need to be re-phrased and explained.

Third, sociological aspects may have an additional or even superior impact on addiction to smoking, which must be referred to.

Author Response

Dear Reviewer,

Thank you very much for your review and valuable comments. We analyzed all the comments and replied to each, indicating where and how the corrections in the Manuscript were made, indicating the line and page.

Below are the point-by-point answers.

With respect

Authors

This is an interesting and brave manuscript where the authors addressed a complex issue involving both genetic and sociological aspects, namely- the relevance of specific polymorphisms in the nicotinic acetylcholine receptors 3 and 5 for the tendency to get addicted to smoking.  The authors reached a well-balanced and careful summary inferring that they are on the right way, but cannot offer a definitive conclusion. This exceptional summation is appreciated and is a positive aspect of this study.

However, the manuscript also includes negative aspects which are weak points and may require corrections in the revised text, as is listed below.

First and foremost, the authors refer to SNPs as if these are the only elements that make a difference in the susceptibility for addiction, while small non-coding RNAs may contribute significantly to the studied phenomenon and were not even mentioned, although others have repeatedly showed significant impact of cholinergic genes-targeting  microRNAs and transfer RNA fragments on diverse related phenomena.

Thank you for this suggestion. A fragment has been added line 68-75.

Second, the issues of sex and gender are presented vaguely, and need to be re-phrased and explained.

Thank You for Your valuable suggestion. We must have really been absent-minded than we forgot to put such an obvious result of our labor. Now it is more complete. This part is inserted in lines 235-238, and a few other authors are referred by the way to make this discussion more interesting and solid.

Third, sociological aspects may have an additional or even superior impact on addiction to smoking, which must be referred to.

Thank You for this note. We tried to do our best to satisfy Your expectations and amplified this subject together with the one above.

Round 2

Reviewer 1 Report

Abstract lines 35-41 are still problematic for several reasons. The paper is not about preclinical models so this doesn't belong. Also, why is this a preliminary report? Having read the paper, it does not look like there are plans to enlarge the study sample. Also it is not clear what is meant that there should be a continuous analysis. 

This whole paragraph could be replaced succinctly with a single sentence that novel haplotypes but with very low frequency variants in CHRNA were associated with nicotine addiction. 

Similarly, lines 408-431 are excessive. The authors should just state again that a common haplotype was not significant. There were two significant haplotypes but the variant frequency was very low, and the findings should be replicated to determine if these have a role in nicotine addiction.  

Line 67 should be modified to "Variants in NachRs" are not only....

line 72 there is a typo

line 270 should be "males were more highly addicted"

line 321, this new sentence is unclear. 

Table 4

Author Response

Dear Reviewer,

Thank you very much for your valuable comments. We analyzed all the comments and replied to each, indicating where and how the corrections in the Manuscript were made, indicating the line and page.

With respect

Author's

Abstract lines 35-41 are still problematic for several reasons. The paper is not about preclinical models so this doesn't belong. Also, why is this a preliminary report? Having read the paper, it does not look like there are plans to enlarge the study sample. Also it is not clear what is meant that there should be a continuous analysis.

This whole paragraph could be replaced succinctly with a single sentence that novel haplotypes but with very low frequency variants in CHRNA were associated with nicotine addiction.

Similarly, lines 408-431 are excessive. The authors should just state again that a common haplotype was not significant. There were two significant haplotypes but the variant frequency was very low, and the findings should be replicated to determine if these have a role in nicotine addiction. 

-Thank you for checking and correcting this section of the article. We changed the conclusions as suggested.

Line: 34-36, page 1 and Line: 370-372, page 9.

The conducted research it is stated that novel haplotypes G-T-T and G-C-T but with very low frequency variants in CHRNA3 were associated with nicotine addiction.

Line 67 should be modified to "Variants in NachRs" are not only....

-Corrected as suggested by the reviewer. line 61 page 2.

line 72 there is a typo

-Corrected as suggested by the reviewer line 65 page 2.

line 270 should be "males were more highly addicted"

-This sentence was adapted as suggested by the reviewer. line 234 page 7.

line 321, this new sentence is unclear.

-The sentence has been deleted.